# Magnetic properties of the putative higher-order topological insulator EuIn$_2$As$_2$

**Tomasz Toliński**[1,3]⋆ **and Dariusz Kaczorowski**[2,3]†

**1** Institute of Molecular Physics, Polish Academy of Sciences,
M. Smoluchowskiego 17, 60-179 Poznań, Poland
**2** Institute of Low Temperature and Structure Research,
Polish Academy of Sciences, Okólna 2, 50-422 Wrocław, Poland
**3** Centre for Advanced Materials and Smart Structures,
Polish Academy of Sciences, 50-422 Wrocław, Poland

⋆ tomtol@ifmpan.poznan.pl

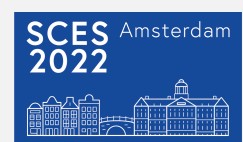

*International Conference on Strongly Correlated Electron Systems
(SCES 2022)
Amsterdam, 24-29 July 2022*

## Abstract

In higher-order topological insulator (HOTI), a gap is preserved both for the bulk and the surface states, and only hinges or corners become gapless. Recently, it has been predicted that the antiferromagnetic compound EuIn$_2$As$_2$ can host both the HOTI and axion insulator features. Preliminary experimental confirmation of this finding was obtained using angle-resolved photoemission spectroscopy. The main objective of this work was to characterize the anisotropic magnetic properties of single-crystalline EuIn$_2$As$_2$. In addition, complementary studies on the transport properties, heat capacity, and magnetocaloric effect in this compound were performed.

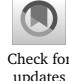

# 1  Introduction

Almost half century old, pioneering works by Kosterlitz, Thouless and Haldane [1, 2] have become a turning point in the creation of new materials that elude classical classification. For example, the archetypal definition of insulators proved unsatisfactory when the strong spin-orbital coupling (SOC) entered the game. Its influence on the valence band has revealed the importance of band topology, as band inversion has become the basis for a class of topological insulators (TI). Here, history has been revived; namely, the nearly 100-year-old idea of Weyl fermions found an unexpected implementation in topological semimetals (TS) that, like the TIs, exhibit unusual surface states - the difference is that the conduction and valence bands join in separated Fermi points. One can discriminate Dirac semimetals (DSMs) and Weyl semimetals (WSMs), the latter resulting from DSM by breaking the space inversion or time reversal symmetry (TRS). TRS can be broken by magnetic order – a feature characteristic of the crystal studied in the current paper. Hence, the role of symmetry breaking is to remove the degeneracy of the Dirac point, which is equivalent to an even number of Weyl points.

Topological materials continue to attract the interest of experimentalists and theoreticians due to the variety of observed quantum phenomena [3–17]. The investigations concentrate on topological insulators and topological semimetals, where the presence of topologically prevented surface or edge states results in fascinating transport properties robust against the typical scattering mechanisms.

The bulk-boundary correspondence in TIs is reflected by gapped bulk states and gapless surface states, the latter protected by the bulk band topology. Unexpectedly, the bulk-boundary correspondence may also be shifted to lower dimensions. In these so-called higher-order topological insulators (HOTI) [18,19], for the 3D TI, the gap is preserved both in the bulk and the surface and only the hinges or corners become gapless. As emphasized by Xie *et al.* [20] HOTIs do not just stem from modification of the boundary states but involve novel topological physics, and new topological invariants become especially relevant.

Our interest is focused on the compound $EuIn_2As_2$, which has been recently claimed [21] to host both the HOTI and axion insulator (AI) features [22–24]. The latter breaks the time reversal symmetry and, moreover, can reveal a presence of 1D chiral states on the hinges. It has been previously suggested that $EuIn_2As_2$ exhibits a canted magnetic order with the intralayer ferromagnetic contribution and antiferromagnetic interlayer exchange [25] like for $EuIn_2P_2$ [26] or like a helical order for $EuMg_2Bi_2$ [27]. Considering the Eu-Eu distances Goforth *et al.* [25] suggested Bloembergen-Rowland or RKKY-type coupling rather than any direct interaction of the Eu cations. Theoretical predictions point to the essential role of magnetic order in the emergence of specific behaviors in this compound. For magnetic moments within the *a-b* plane it can be a topological crystalline insulator (TCI), characterized by nonzero mirror Chern numbers [21] and for easy magnetization direction along the *c* axis, it is predicted to be a HOTI. Moreover, neutron diffraction and symmetry analysis carried out by Riberolles *et al.* [28] have

indicated that EuIn$_2$As$_2$ is a magnetic topological-crystalline axion insulator protected by 2' symmetry. They also suggested that this system can reveal either gapped or gapless surface states depending on the direction of the applied magnetic field, therefore, dependent on the magnetic order. These neutron diffraction studies have shown that the magnetic structure is composed of ferromagnetic layers within the hexagonal *a-b* plane, which are rotating helically along the *c* axis. More interestingly, they have found a narrow range below $T_N$, were pure 60°-helix order exists followed by a broken-helix order.

Next, it stems from the thorough angle-resolved photoemission spectroscopy (ARPES) measurements and first-principles calculations by Regmi *et al.* [13] that the in-plane and out-of-plane configurations are energetically nearly degenerate and both are characterized by topological invariant $\mathbb{Z}_4 = 2$, supporting the axion insulator state.

Considering the above predictions, in the present studies we aimed at comprehensive characterization of the thermodynamic (magnetization, heat capacity, magnetocaloric effect) and electrical transport (resistivity, magnetoresistance) properties of high-quality single crystals of EuIn$_2$As$_2$. The results of our measurements, performed in wide ranges of temperature and magnetic field strength, allowed to construct the magnetic phase diagrams for the two principal directions in the hexagonal unit cell of the compound.

## 2 Experimental

Single crystals of EuIn$_2$As$_2$ were grown from In-As flux. High purity constituent elements (purity: Eu 3N, In 5N, As 5N) were taken in a molar ratio Eu:In:As of 1:12:3. The synthesis was carried out using an **ACP-CCS-5** Canfield Crucible Set (LCP Industrial Ceramics Inc.) sealed in an evacuated quartz tube. The mixture was heated up to 1000°C at a rate of 50°C/h, maintained at this temperature for 24 hours, and then slowly cooled down to 700°C at a rate 2°C/h. Subsequently, the ampule was removed from the furnace, flipped over and centrifuged in order to remove the flux mixture. As a product of the entire procedure, several silver-shiny platelet-like crystals were obtained (see Fig. 1) with dimensions up to 2 × 2×0.1 mm. They were found to be stable in air and moisture. As a byproduct, several well-developed InAs crystals were grown.

Chemical composition and phase homogeneity of the prepared crystals were determined by energy-dispersive X-ray (EDX) analysis performed using a FEI scanning electron microscope equipped with an EDAX Genesis XM4 spectrometer. In order to verify the crystal symmetry of the obtained single crystals, a small fragment was crushed from a larger piece and examined on an Oxford Diffraction X'calibur four-circle single-crystal X-ray diffractometer equipped with a CCD Atlas detector. Crystallinity and crystallographic orientation of the particular crystals used in physical measurements were determined by means of Laue X-ray backscattering technique implemented in a LAUE-COS (Proto) system.

Magnetic properties were investigated in the temperature range 1.72 – 300 K and in magnetic fields up to 9 T using a Quantum Design MPMS-XL superconducting quantum interference device (SQUID) magnetometer. In addition, the magnetization measurements were carried out in the interval 1.9 – 400 K and in fields up to 9 T using a Quantum Design PPMS-9 platform equipped with a vibrating sample magnetometer (VSM). Heat capacity was measured from 1.9 K to 300 K and in magnetic fields up to 9 T employing the relaxation technique and the two-$\tau$ method implemented in the same PPMS-9 platform. Electrical transport studies were carried out in the temperature interval 2 – 300 K and in fields up to 9 T using a Quantum Design PPMS-14 platform. The electrical leads were made of gold wires attached to the bar-shaped specimens with silver-epoxy paste. The experiments were done employing a standard four-point ac technique. In the magnetoresistance measurements, the magnetic field was applied

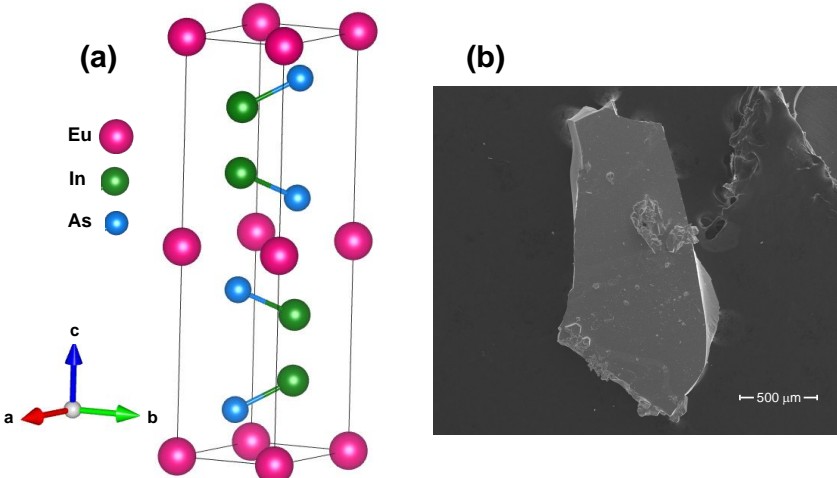

Figure 1: (a) Crystal structure of EuIn$_2$As$_2$. Only selected bonds between In and As ions are displayed to better visualize the layered nature of the structure. (b) Exemplary piece of single crystal, used for EDAX measurements with the obtained stoichiometry (At%): Eu: 20.18, In: 39.92, As: 39.89.

in direction perpendicular the *a-b* plane.

## 3 Structural and microstructural characterization

The X-ray diffraction experiments on EuIn$_2$As$_2$ yielded a hexagonal unit cell (space group $P6_3/mmc$) with the lattice parameters $a = 4.207$ Å and $c = 17.889$ Å, in good agreement with the literature data [25, 29].

Characteristic for this structure is a layered arrangement along the crystallographic *c* axis, composed of alternating anion layers of $[In_2As_2]^{2-}$ and cation layers of $Eu^{2+}$. In Fig. 1a the crystal structure of EuIn$_2$As$_2$ is presented, whereas Fig. 1b shows exemplary piece of crystal used to determine the stoichiometry using EDAX. The EDX results proved single-phase character of the specimens. Calculations of the stoichiometry for a few pieces of the synthesized EuIn$_2$As$_2$ crystals using EDAX showed a good agreement with the nominal composition.

## 4 Magnetic properties

Figure 2 presents the magnetic susceptibility of single-crystalline EuIn$_2$As$_2$, measured in the temperature range 2-400 K with two principal magnetic field orientations, i.e., *H* confined within the hexagonal *a-b* plane and *H* applied along the *c* axis. The measurements were carried out in zero-field-cooling (ZFC) and field-cooling (FC) modes. In the entire temperature range, the two curves were found to overlap, so in the figure only the ZFC data is shown. The compound exhibits antiferromagnetc (AFM) ordering below the Néel temperature $T_N = 16.1(1)$ K, in good agreement with the previous magnetometric studies [25, 29]. In the paramagnetic state, up to about 30 K, one observes a small difference in the magnetic susceptibility measured with $H \parallel ab$ and $H \parallel c$, and this effect likely arises due to short-range interactions. At higher temperatures, the magnetic susceptibility becomes isotropic, as expected for a material with no crystalline electric field effect ($Eu^{2+}$ is a spin-only ion). The magnetic data obtained on a set of several randomly oriented single crystals is plotted in the inset to Fig. 2 in a form

**SciPost** | SciPost Phys. Proc. 11, 005 (2023)

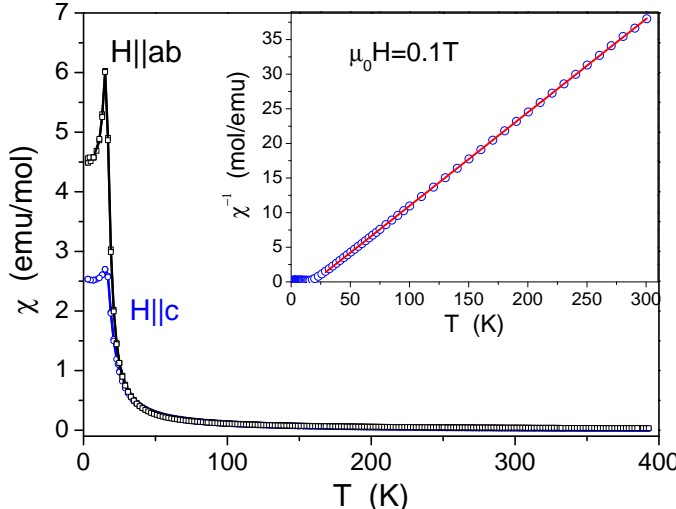

Figure 2: Temperature dependencies of the magnetic susceptibility of EuIn$_2$As$_2$ measured in a magnetic field $\mu_0 H = 0.1$ T aligned along the crystallographic $c$ axis and within the hexagonal $a$-$b$ plane, upon cooling the specimen to 2 K in zero field (ZFC mode). Inset: Inverse magnetic susceptibility of EuIn$_2$As$_2$ measured on a set of randomly oriented crystals. The solid red line represents the CW fit (see the text).

of the reciprocal magnetic susceptibility. Fitting this data with the Curie–Weiss (CW) law

$$\chi^{-1} = \frac{3k(T - \theta_{\mathrm{p}})}{N_A \mu_{\mathrm{eff}}^2} \tag{1}$$

yielded the paramagnetic Curie temperature $\theta_{\mathrm{p}} = 18.7(1)$ K, and the effective magnetic moment $\mu_{\mathrm{eff}} = 7.7(1)\ \mu_{\mathrm{B}}$. The latter value is close to the theoretical effective magnetic moment of Eu$^{2+}$ ion ($g\sqrt{J(J+1)} = 7.94$; Lande factor $g = 2$, total angular momentum equal to spin angular momentum $J = S = 7/2$). The large positive value of $\theta_{\mathrm{p}}$ hints at the predominance of ferromagnetic (FM) magnetic exchange interactions in the AFM compound.

Fig. 3a shows the magnetization isotherms EuIn$_2$As$_2$ measured at $T = 2$ K with $H \parallel ab$ and $H \parallel c$. The magnetization process along the $c$ axis reveals a continuous rotation of the magnetic moments with increasing magnetic field, followed by a saturation above the critical field $\mu_0 H_{\mathrm{sat}}^{\mathrm{c}} = 2.0$ T at a value of 6.9 $\mu_{\mathrm{B}}$ that is very close to the theoretical prediction for divalent Eu ion ($gJ = 7.0$). In the case of $H \parallel ab$, the saturated magnetic moment is the same, however the critical field is significantly smaller being $\mu_0 H_{\mathrm{sat}}^{\mathrm{ab}} = 0.9$ T. As can be inferred from Fig. 3b, the two field dependencies of the magnetization notably differ in a weak magnetic fields region. While that obtained with $H \parallel c$ is linear in $H$, the $H \parallel ab$ isotherm shows first a coherent rotation of the moments in the magnetic field interval 0 – 0.15 T, and then a metamagnetic-like transition in the range 0.15 – 0.3 T, followed by another region of smooth rotation of the moments observed up to $\mu_0 H_{\mathrm{sat}}^{\mathrm{ab}}$. Remarkably, the nonlinear behavior bears a hysteretic character which points to a spin-flop transition. Altogether, the magnetization data indicates that the crystallographic $c$ axis represents a hard magnetic direction in EuIn$_2$As$_2$, and the europium magnetic moments are confined in this compound within the $a$-$b$ plane.

In order to shed more light on the character of the magnetic interactions in EuIn$_2$As$_2$, the low-temperature magnetic susceptibility was measured in various external magnetic fields, and the results are presented in Fig. 4. In small magnetic fields, $\chi_{\mathrm{c}}(T)$ in the AFM state is not only much smaller than $\chi_{\mathrm{ab}}(T)$ but also almost temperature independent, in line with the easy-plane magnetism. For both main field directions, the anomaly denoting $T_N$ shifts toward

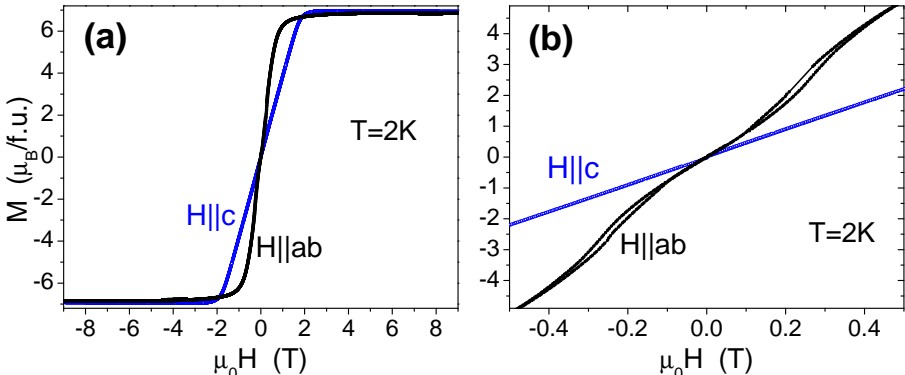

Figure 3: (a) Magnetic field variations of the magnetization in single-crystalline EuIn$_2$As$_2$ measured at $T = 2$ K in magnetic fields applied along the $c$ axis and in the $a$-$b$ plane. (b) Magnification of the magnetization isotherms in a weak fields region.

lower temperatures and gradually broadens as the field strength increases, as expected for AFM systems. The metamagnetic-like transition near 0.25 T occurring for $H \parallel ab$ (see Fig. 4a) manifests itself as an additional peak, which together with the main magnetic transition creates a double-kink feature in $\chi_{ab}(T)$ taken in magnetic fields of 0.15 T and 0.2 T. Panels (c)-(f) in Fig. 4 illustrate the methodology used to determine the characteristic temperatures, $T_N$ and $T_{SF}$, based on the derivatives of the magnetic susceptibility.

Further, Fig. 5 shows the magnetization isotherms $M(H)$ measured densely in the temperature range 4 - 40 K. They reveal the expected behavior, i.e. a gradual change from the FM-like saturation in strong magnetic fields at the lowest temperatures to a tendency to such saturation at temperatures close to $T_N$ to a Brillouin-type functional dependence at high temperatures. Relatively significant curvature of the $M(H)$ variations at $T > T_N$ can be attributed to strong short-range interactions (see below). The numerous magnetization curves with progressive change of the slope near $H = 0$ allowed to determine the zero-field susceptibility (not shown), which perfectly confirmed the ordering temperature $T_N = 16.1(1)$ K. Remarkably, while the neutron diffraction studies by Riberolles *et al.* [28] have shown two AFM transitions, at $T_{N1} = 17.6(2)$ K and $T_{N2} = 16.2(1)$ K, in the present study only the latter one could be detected. However, we do not definitely exclude the presence of the transition at $T_{N1}$, which can be masked by overlapping anomalies associated with the various transitions at $T_{N1}$, $T_{N2}$, and $T_{SF}$. The occurrence of the transition at $T_{N1}$ or its absence may also depend on subtle differences in the crystal structure, such as atom disorder, deviations from ideal stoichiometry and/or tiny chemical doping.

The data of Fig. 5 can be also displayed in a form of Arrott plots around the Néel temperature $T_N = 16.1(1)$ K (Fig. 6). The main difference between the curves obtained for the $H \parallel ab$ and $H \parallel c$ orientations is a negative curvature of the $H \parallel ab$ plots seen in the ordered state in weak magnetic fields (note lower panels of Fig. 6). According to the Banerjee criterion [30], this feature is a characteristic signature of first-order phase transitions, and in the present case it corresponds to the spin-flop transition observed for the easy magnetization plane.

## 5 Magnetocaloric effect

Based on the magnetization isotherms collected in Fig. 5, one can derive the temperature dependencies of the magnetocaloric effect (MCE). The isothermal magnetic entropy change

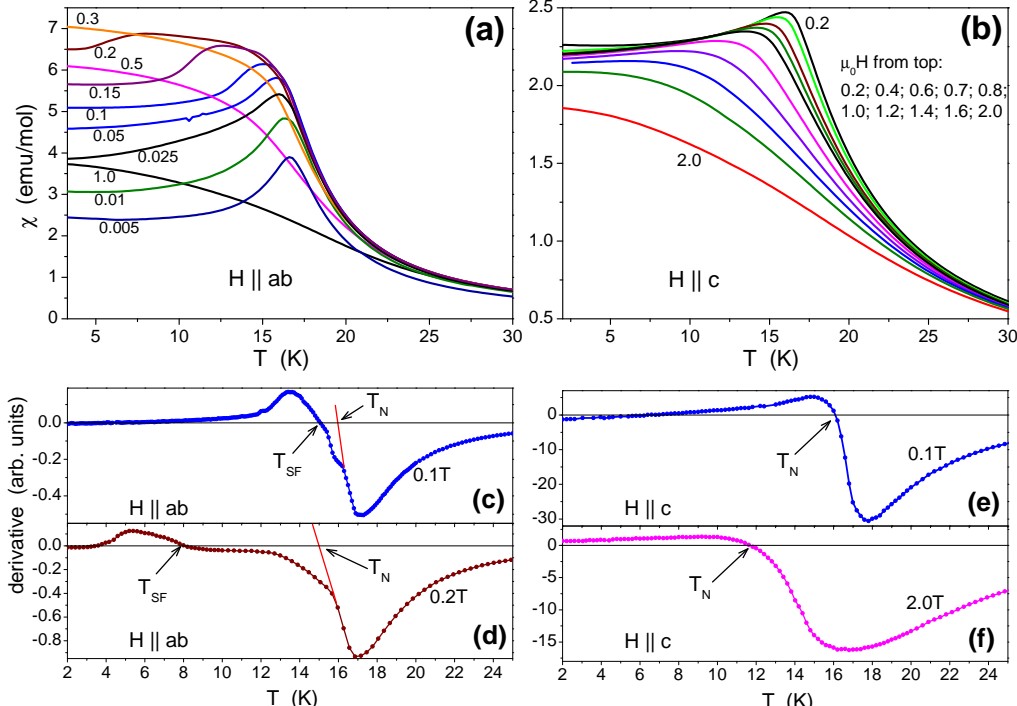

Figure 4: Low-temperature dependencies of the magnetic susceptibility of single-crystalline EuIn$_2$As$_2$ measured in several different magnetic fields oriented (a) within the $a$-$b$ plane, and (b) along the $c$ axis. (c)-(f) Examples of the determination of the characteristic temperatures based on the derivatives of the magnetic susceptibility.

can be determined from the formula

$$\Delta S_{\mathrm{M}} \approx \frac{\mu_0}{\Delta T}\left[\int_0^{H_{\max}} M(T+\Delta T,H)dH - \int_0^{H_{\max}} M(T,H)dH\right], \qquad (2)$$

where $\Delta T$ represents difference between temperatures of two magnetization curves, $dH$ stends for a magnetic field step, and $M(T,H)$ and $M(T+\Delta T,H)$ are values of the magnetization at temperatures $T$ and $T+\Delta T$, respectively. Figures 7a and 7b display $\Delta S_{\mathrm{M}}(T)$ for the two field orientations considered. The magnitude of the entropy change near $T_{\mathrm{N}}$ is fairly large, similar to the data reported for good themoelectrics with comparable ordering temperatures [31]. In particular, it is very close to the result obtained for EuFe$_2$As$_2$ [32].

Closer inspection of $\Delta S_{\mathrm{M}}(T)$ derived for small changes of the magnetic field (see Fig. 7c and Fig. 7d) shows that for $H \parallel ab$ an additional contribution characterized by negative values is visible at low temperatures. This contribution disappears progressively with increasing the magnetic field strength and smears out above about 0.3 T. The latter field value coincides with the spin-flop transition. The observed behavior likely stems from the predominance of the FM interactions in the hexagonal $a$-$b$ planes. In turn, for $H \parallel c$, a similar negative contribution at around 10 K is visible in fields up to $\mu_0 H_{\mathrm{sat}}^{\mathrm{c}} = 2.0$ T, and can be associated with the AFM interactions between the $a$-$b$ layers of the Eu ions.

Presumably, the strong FM interactions active within the $a$-$b$ planes are responsible for larger magnitude of the magnetic entropy change for the $H \parallel ab$ orientation (see Fig. 8a). However, the rotating magnetocaloric effect (R-MCE), defined as $\Delta S_{\mathrm{diff}} = \Delta S_c - \Delta S_{ab}$, achieves in a field of 5 T rather moderate values, and it is much smaller than giant R-MCE effect reported recently for TbScO$_3$ (23.6 J/kgK) [33].

A useful parameter describing magnetocaloric performance is the relative cooling power

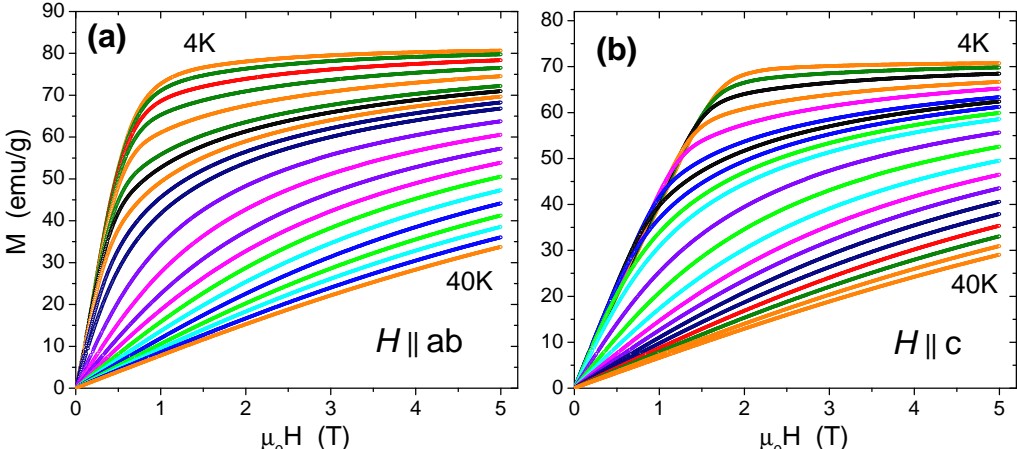

Figure 5: Magnetization isotherms of single-crystalline $EuIn_2As_2$ measured at several different temperatures in magnetic fields oriented (a) within the *a-b* plane, and (b) along the *c* axis.

(RCP) defined as a product of the maximum value of $\Delta S_M$ and the full width at half maximum of the $\Delta S_M(T)$ peak:

$$RCP = \left| -\Delta S_M^{max}(T,H) \right| \times \delta T_{FWHM}. \qquad (3)$$

As can be inferred from Fig. 8b, the magnitude of RCP derived for $EuIn_2As_2$ is rather small, mainly because of narrowness of the peaks in the $\Delta S_M(T)$ dependencies. Other systems showing similar values of $\Delta S_M$ at comparable ordering temperatures, for example $DyCo_3B_2$ [34], exhibit larger values of RCP. However, an advantage of $EuIn_2As_2$ is the lack of hysteresis in the $M(H)$ variations that limits the losses in magnetocaloric applications.

# 6 Specific heat

Figure 9 shows the temperature dependence of the specific heat of $EuIn_2As_2$. Near room temperature, the $C_p(T)$ curve tends to saturate at a value of 122.8 J $mol^{-1}K^{-2}$ that is very close to the Dulong–Petit limit $3nR$ ($n = 5$ stands for a number of atoms in the formula unit and $R = 8.31$ J $mol^{-1}K^{-1}$ is the universal gas constant). In the paramagnetic state, the experimental data can can be well described by the standard function that takes into account Debye and Einstein phonon modes, characterized by the characteristic temperatures $\theta_D$ and $\theta_E$, respectively:

$$C_p(T) = \gamma T + nR\left[(1-w)9\left(\frac{T}{\theta_D}\right)^3 \int_0^{x_D} \frac{x^4 exp(x)}{(exp(x)-1)^2}dx + 3w\frac{x_E^2 exp(x_E)}{(exp(x_E)-1)^2}\right], \qquad (4)$$

where $x_D = \theta_D/T$, $x_E = \theta_E/T$, $\gamma$ stands for the electronic specific heat coefficient, and $w$ is a weight parameter of the Debye and Einstein contributions. Fitting the experimental data with this function yielded: $\gamma = 13.6(9)$ $mJmol^{-1}K^{-2}$, $\theta_D = 348(3)$ K, $\theta_E = 107.4(8)$ K, and $w = 0.5$. The analysis was carried out above 35 K, i.e. in the region of negligible short-range interactions, evident from the magnetic susceptibility studies. The obtained values of the parameters are similar to those reported for isostructural Eu-based systems (e.g., for $EuMg_2Sb_2$ they are $\theta_D = 415$ K, $\theta_E = 111$ K, and $w = 0.48$ [35]).

The AFM phase transition in $EuIn_2As_2$ manifests itself as a distinct mean-field-like anomaly in $C_p(T)$ at $T_N = 16.1$ K. Its evolution in magnetic field applied along the hexagonal *c* axis

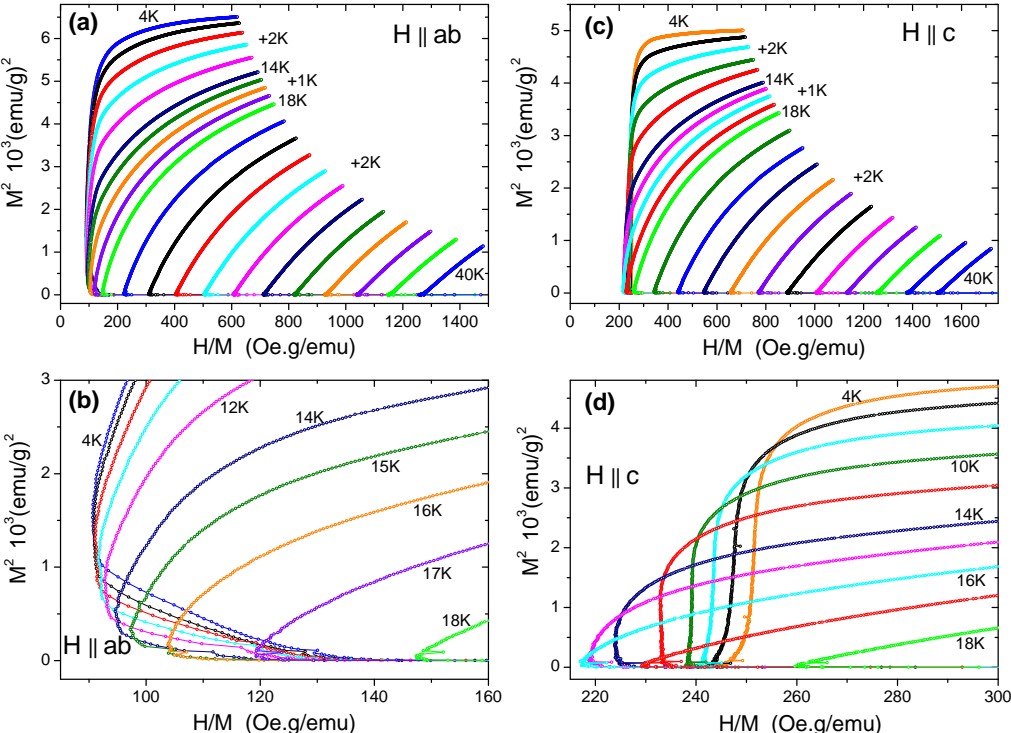

Figure 6: Arrott plots made for single-crystalline EuIn$_2$As$_2$ in the temperature range 4-40 K and magnetic fields up to 5 T applied (a) within the *a-b* plane, and (c) along the *c* axis. Panels (b) and (d), respectively, show the same data constrained to the temperature range 4-18 K, i.e. in the vicinity of the AFM phase transition.

is visualized in Fig. 10. With increasing the magnetic field strength, the peak gradually shifts towards lower temperatures and its amplitude diminishes, as expected for antiferromagnets. In fields stronger than $\mu_0 H^c_{\text{sat}} = 2.0$ T, no singularity corresponding to the AFM phase transition is seen, as the compound is in the field-induced FM state, revealed in the magnetic studies. As can be inferred from the inset in Fig. 10, within experimental accuracy no anomaly other than that related to the AFM ordering at 16.1 K can be detected in the heat capacity data unambiguously neither in zero nor finite magnetic field.

The magnetic contribution to the specific heat below $T_N$ was derived by subtracting from the measured $C_p(T)$ data the phonon contribution calculated from Eq. 4. The so-obtained $C_{\text{mag}}(T)$ variation is shown in the inset to Fig. 11, together with $C_{\text{mag}}(t)$ calculated (without any free parameters) from the formula [36]:

$$C_{\text{mag}}(t) = R \frac{3S\overline{\mu}_0^2(t)}{(S+1)t\left[\tau^*(t) - 1\right]}, \tag{5}$$

where the reduced parameters are:

$$t = \frac{T}{T_N}, \quad \tau^*(t) = \frac{(S+1)t}{3B'_S(y_0)}, \quad y_0 = \frac{3\overline{\mu}_0}{(S+1)t}, \quad \overline{\mu}_0 = \frac{\mu_0}{gS\mu_B} = B_S(y_0), \tag{6}$$

and $\mu_0$ is a zero-field magnetic moment calculated within the Brillouin function $B_S(y_0)$. The forms of the latter function and its derivative are given in Refs. [36, 37]. From comparison of the model and the measured $C_{\text{mag}}(T)$ it is clear that the experimental data forms a tail above $T_N$, extending up to about 30 K, which manifests the strong short-range interactions. In the AFM state, the theoretical description gives more pronounced hump in $C_{\text{mag}}(T)$ that

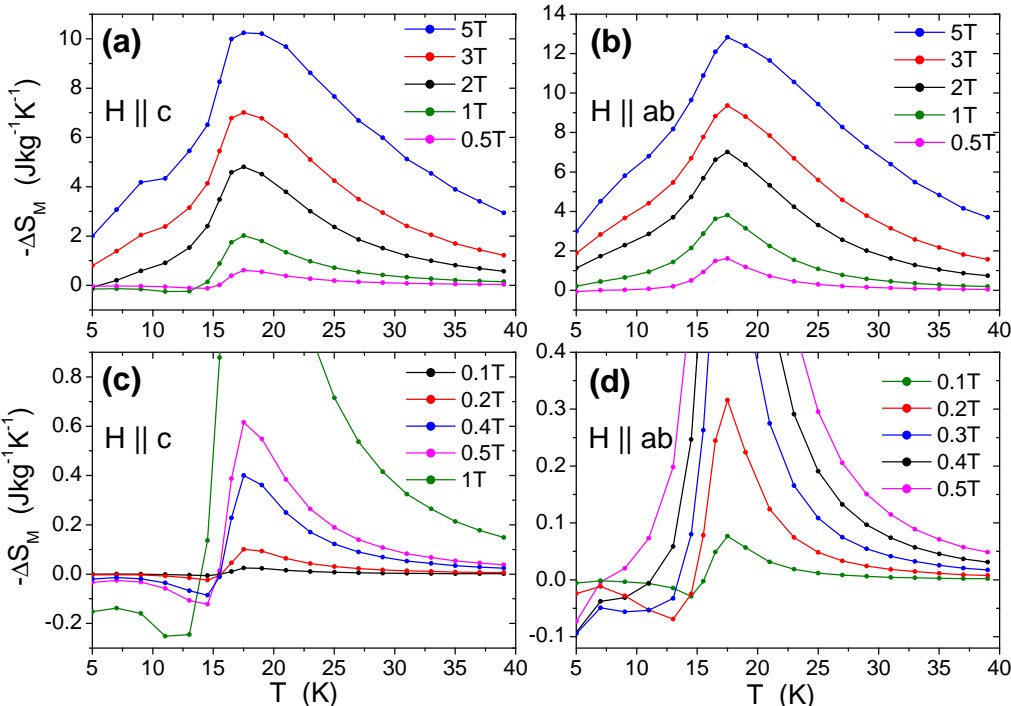

Figure 7: Temperature dependencies of the magnetocaloric effect in EuIn$_2$As$_2$ estimated for different strength of the magnetic field applied (a,c) along the $c$ axis, and (b,d) within the $a$-$b$ plane.

arises due to Zeeman splitting of the $J = 7/2$ multiplet, typical of Eu$^{2+}$ systems. The observed discrepancy brings about a deficiency in the magnetic entropy released by $T_N$, that must be shifted to the paramagnetic region.

By integrating the $C_{mag}/T(T)$ curve one can derive the temperature dependence of the magnetic entropy, and this result is displayed in the main panel of Fig. 11. At $T_N = 16.1(1)$ K, the entropy equals 12.7 J/molK$^2$, which is ca. 73% of the value $R\ln(2S+1) = 17.3$ J/molK$^2$, expected for Eu$^{2+}$ ion. The missing entropy can be associated with the afore-mentioned short-range interactions. The $S_{mag}(T)$ variation saturates at the theoretical value above about 30 K, i.e. at temperatures where neither magnetic susceptibility nor specific heat are affected by the short-range order effect.

## 7 Electrical magnetotransport

The temperature dependence of the electrical resistivity of EuIn$_2$As$_2$ single crystal measured with electric current flowing in the hexagonal $a$-$b$ plane is shown in Fig. 12. Both the magnitude of the resistivity and the shape of $\rho(T)$ indicate a semimetallic character of the compound, in line with the previous reports [25, 38]. With decreasing temperature from 300 K, the resistivity decreases in an almost linear manner but near 100 K $\rho(T)$ forms a broad minimum and at lower temperatures it exhibits a distinct upturn observed down to the onset of the AFM state.

The AFM transition in EuIn$_2$As$_2$ manifests itself as a pronounced maximum in the temperature derivative of $\rho(T)$. The so-determined $T_N$ equals 16.2 K, in perfect agreement with the thermodynamic data. In the ordered state, the resistivity sharply decreases due to reduced spin disorder scattering, and can be approximated by the formula [39]:

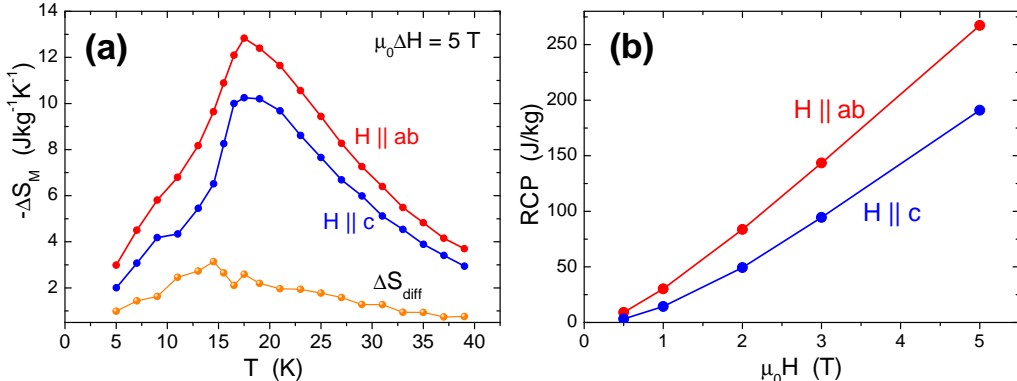

Figure 8: (a) Comparison of the magnetocaloric effect in EuIn$_2$As$_2$ for a magnetic field change of 5 T and field applied parallel to the *c* axis and within the *a-b* plane. The bottom curve represents the rotating MCE effect discussed in the text. (b) Comparison of the relative cooling power derived for EuIn$_2$As$_2$ as a function of magnetic field for the $H \parallel ab$ and $H \parallel c$ orientations.

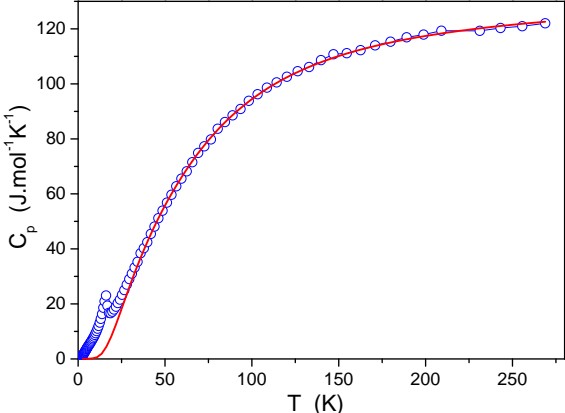

Figure 9: Temperature variation of the specific heat of single-crystalline EuIn$_2$As$_2$. The solid red curve is the Debye-Einstein fit described in the text.

$$\rho(T) = \rho_0 + B\Delta_{\mathrm{mag}}^2 \sqrt{\frac{T}{\Delta_{\mathrm{mag}}}} e^{-\Delta_{\mathrm{mag}}/T} \left[ 1 + \frac{2}{3}\frac{T}{\Delta_{\mathrm{mag}}} + \frac{2}{15}\left(\frac{T}{\Delta_{\mathrm{mag}}}\right)^2 \right], \tag{7}$$

where $\rho_0$ is the residual resistivity and the second term accounts for scattering conduction electrons on antiferromagnetic magnons ($\Delta_{\mathrm{mag}}$ denotes a gap in the AFM spin waves spectrum). The parameters obtained from fitting Eq. 7 to the experimental data of EuIn$_2$As$_2$ are: $\rho_0 = 0.18$ mΩ cm, $B = 1.1 \times 10^{-3}$ mΩ cm K$^{-2}$, and $\Delta_{\mathrm{mag}} = 4.7$ K. The parameter $B$ is a material-constant, which is related to the spin wave velocity and the obtained value is of typical order.

The upturn in $\rho(T)$ observed above the maximum resembles the behaviour reported in similar Eu-based compounds [26, 40, 41], which has been often attributed to a metal-semiconductor transition (see, e.g., Ref. [26]). Considering the short-range order, clearly observed in the magnetic data, a different mechanism can be considered, namely trapping magnetic polarons (MP) [38, 42]. In this scenario, the electrical transport just above $T_{\mathrm{N}}$ is governed by charge mobility of hopping type. As can be inferred from the inset to Fig. 12, the $\rho(T)$ variation in the interval 20 – 100 K can be described by the function

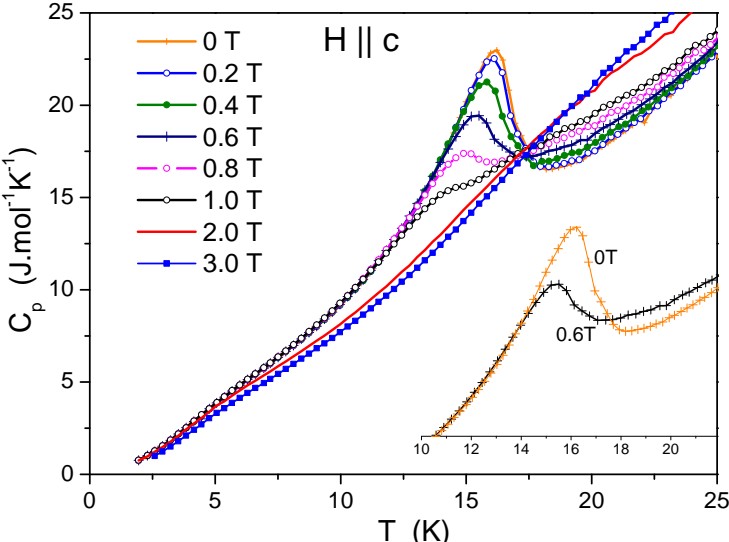

Figure 10: Low-temperature dependencies of the specific heat of single-crystalline EuIn$_2$As$_2$ measured in different magnetic fields applied along the hard magnetization axis. Inset: exemplary data in the vicinity of the AFM phase transition.

$$\rho(T) = a_1 + a_2 T + a_3 exp\left[\left(\frac{T_M}{T}\right)^{1/4}\right], \tag{8}$$

where the first term is a sum of temperature independent contributions (residual resistivity due to structural defects, scattering conduction electrons on disordered magnetic moments), the second term represents electron-phonon scattering processes approximated by high-temperature limit of the Bloch-Grüneissen function, and the third term accounts for MP hopping expressed in a Mott variable-range hopping form ($T_M$ is a characteristic temperature that measures an average energy level spacing) [43]. The parameters obtained from fitting Eq. 8 to the experimental data of EuIn$_2$As$_2$ are: $a_1 = 0.18$ mΩ cm, $a_2 = 9.4 \times 10^{-4}$ mΩ cm K$^{-1}$, $a_3 = 0.01$ mΩ cm, and $T_M = 2860$ K. It can be noticed that the value of $a_1$ equals to $\rho_0$ estimated from the data measured in the AFM state, which implies that the spin-disorder scattering provides a minor contribution to the total resistivity measured above $T_N$.

Fig. 13 displays the magnetic field dependencies of the transverse magnetoresistance (MR) of EuIn$_2$As$_2$, defined as $\Delta\rho/\rho(H) = [\rho(H) - \rho(H=0)]/\rho(H=0) \times 100\%$. The measurements were performed at a few temperatures in the range 0.4 K – 20 K for electric current flowing in the crystallographic $a$-$b$ plane and magnetic field directed along the $c$ axis. Below 12 K, $\Delta\rho/\rho(H)$ undergoes first through a minimum located near 0.8 T and then forms a maximum at about 1.7 T, i.e. slightly below the critical field $\mu_0 H_{sat}^c$. In the entire field range, MR is negative, and in fields stronger than $\mu_0 H_{sat}^c$ it bears a shape characteristic for ferromagnets. At higher temperatures, no minimum in $\Delta\rho/\rho(H)$ is seen, and the MR maximum becomes positive and gradually shifts towards smaller fields with increasing temperature. The latter behavior is reminiscent to that typical of antiferromagnets near a metamagnetic transition. Slightly above $T_N$ (note in Fig. 13 the isotherm taken at $T = 20$ K), MR is negative and its magnitude smoothly increases with increasing the magnetic field strength, in concert with the dominant FM character of the short-range interactions in this temperature region. It is worth noting, that MR measured at 20 K attains as large a value as -62% in a magnetic field of 9 T.

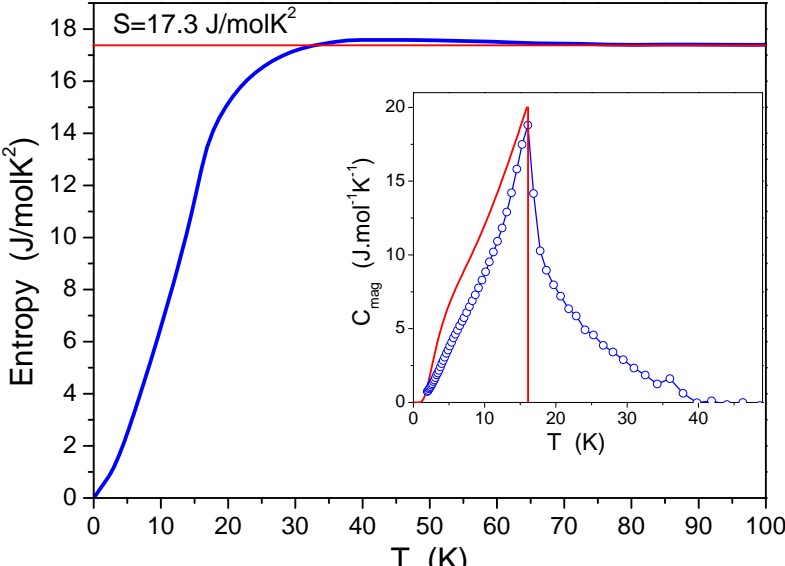

Figure 11: Temperature dependence of the magnetic entropy in EuIn$_2$As$_2$. Inset: Low-temperature variation of the magnetic contribution to the specific heat (circles) and the theoretical description within the mean field model (red lines).

# 8 Magnetic phase diagrams

The experimental data collected for single-crystalline EuIn$_2$As$_2$ can be used in constructing the magnetic phases diagrams, which represent the magnetic properties of the crystals subjected to external magnetic field directed within the crystallographic *a-b* plane (see Fig. 14a) and along the hexagonal *c* axis (see Fig. 14b). In both configurations, the AFM phase transition was identified from a maximum (or high-temperature inflection point) in the $\chi(T)$ curves taken in different magnetic fields (Fig. 4) as well as from a knee in the field derivative of the magnetization $M(H)$ isotherms measured at different temperatures (Fig. 5). Fig. 15 illustrates an example of the latter approach, where the critical fields $H_{cr}$ set points on the $T_N$ versus $H$ dependencies presented in Fig. 14. In the case of magnetic field applied within the easy magnetization plane, an additional order-order boundary line exists that can be derived from the $\chi(T)$ data (note in Fig. 4a the low-temperature anomalies seen in the field range 0.1 – 0.3 T) as well as from the $M(H)$ variations given in Fig. 5a (as an example, note a sharp peak at $H_{sf}$ in the d$M$/d$H(H)$ isotherm shown in Fig. 15a). As the associated spin-flop (SF) transition is hysteretic in nature (see the inset in Fig. 3b), the phase boundary was identified as a midpoint in the anomaly in $M(H)$. Both magnetic phase diagrams of EuIn$_2$As$_2$ have yet another line that marks a transition into fully polarized state. This phase boundary was derived from the d$M$/d$H(H)$ curves (note $H_{sat}$ in Fig. 15), and for $H \parallel c$, additionally from the MR data (see Fig. 13b).

The neutron diffraction experiment of Riberolles *et al.* [28] revealed that below $T_{N2} = 16.1$ K the magnetic order in EuIn$_2$As$_2$ has a form of broken-helix, for which the turn angle is larger than 60° and systematically increases with lowering the temperature reaching a value of about 130° at $T \rightarrow 0$. Upon applying magnetic field along the hard magnetic direction, a fan-type structure of the Eu magnetic moments is created, and an increase in the field strength is accompanied with a gradual rise in the spin component projected on the *c* axis. At the same time, the ordering temperature gradually decreases. At a given temperature $T < T_N$, crossing the AFM phase boundary set by $H_{cr}$ brings about breakdown in the interplane antiferromagnetic coupling, and entering into a field-induced state in which *a-b* layers of FM

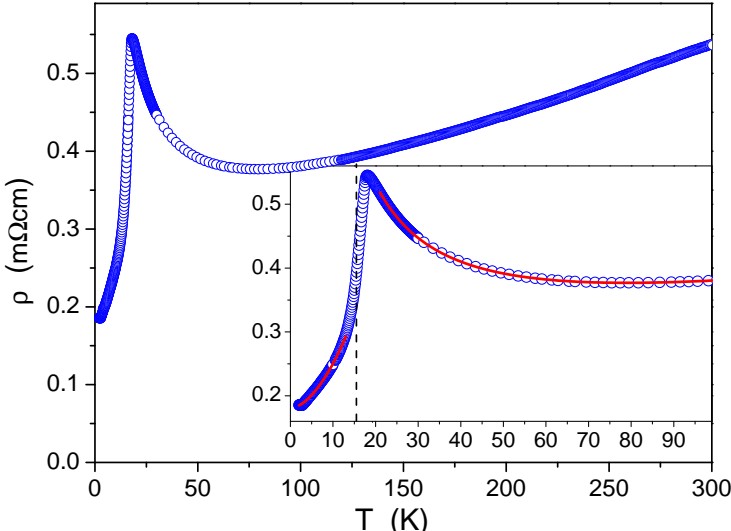

Figure 12: Temperature dependence of the electrical resistivity of single-crystalline EuIn$_2$As$_2$ measured with electric current flowing in the *a-b* plane. The inset displays the low-temperature resistivity data, and their least-squares fits (solid red lines) discussed in the text.

coupled magnetic moments progressively orient towards the field direction. Eventually, in fields stronger than $H_{sat}$, a fully polarized phase is achieved.

For magnetic field applied along the easy magnetic plane, the broken-helix structure is first transformed via the SF transition to a phase with magnetic moments oriented perpendicular to the field direction. With further rise in the field strength the spins rotate towards the magnetic field direction until the breakdown of the AFM coupling. In stronger fields, a fan structure is formed, made by the magnetic moments confined in the *a-b* planes. At the critical field $H_{sat}$, a fully polarized state is realized in which all the spins point towards the magnetic field direction. It is worth noting, that the value of $H_{sat}$ found for this field orientation is significantly smaller than that observed for $H \parallel c$, where external magnetic field must additionally overcome the magnetic anisotropy field.

## 9 Summary

We performed comprehensive studies on high quality single crystals of EuIn$_2$As$_2$ employing magnetic, thermal, and transport measurements. Such an extended characterization of this compound is of special importance as it can create a powerful platform for research on different topological states. It results from the prediction that EuIn$_2$As$_2$ can demonstrate both HOTI and AI behaviour depending on the actual magnetic state.

The compound was found to order antiferromagnetically below $T_N = 16.1(1)$ K, and to exhibit sizeable short range interactions of ferromagnetic character in the paramagnetic state observed up to about 35 K. The magnetism in EuIn$_2$As$_2$ is due to divalent Eu ions with $4f^7$ electronic configuration characterized by the spin-only $_8S^{7/2}$ state. The magnetic structure consists of ferromagnetically coupled *a-b* layers stacked antiferromagnetically along the crystallographic *c* axis. Based on the temperature and magnetic field dependencies of the magnetization, specific heat, and electrical resistance we constructed the magnetic phase diagrams for the two main orientations of external magnetic field, i.e., for $H$ applied within the hexagonal plane and along the *c* axis, which correspond to the easy magnetic plane and the hard magne-

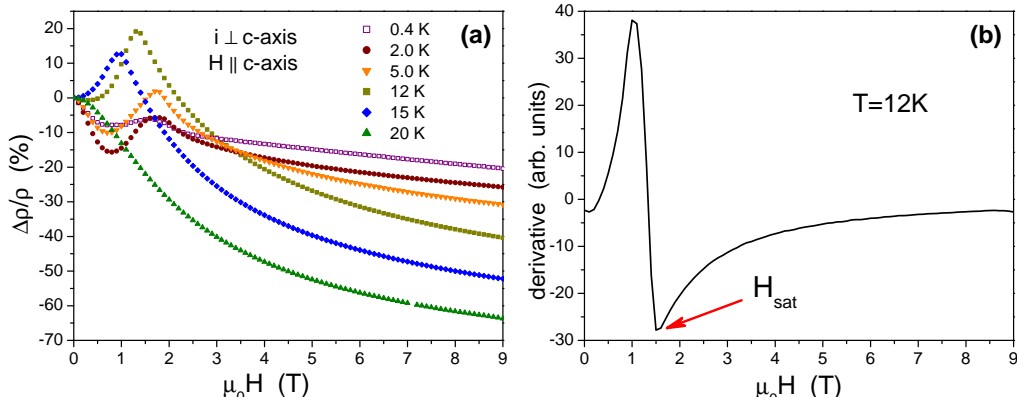

Figure 13: (a) Transverse magnetoresistance isotherms measured on single crystalline $EuIn_2As_2$ with magnetic field aligned along the hexagonal $c$ axis and electric current flowing in the $a$-$b$ plane. (b) Magnetic field derivative of the magnetoresistance isotherm taken at 12 K, as an example of the determination of the critical field $H_{sat}$ above which the full magnetic saturation state is observed (see the text).

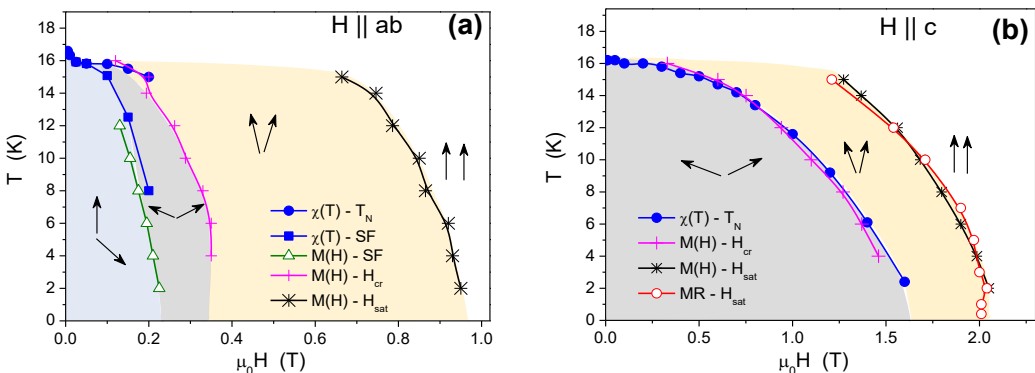

Figure 14: Magnetic phase diagrams of single-crystalline $EuIn_2As_2$ built for (a) $H \parallel ab$ and (b) $H \parallel c$. Meaning of the symbols is given in the figures, and their derivations are explained in the main text. Different magnetic phases are marked by different colours and by schematic arrangements of magnetic moments with respect to the field direction.

tization direction, respectively. The obtained data indicates that by changing the orientation and strength of the applied magnetic field, one can stabilize in $EuIn_2As_2$ different magnetic orders, such as broken-helix, fan type, and field-induced fully polarized ones [44]. This finding corroborates the previous experimental results [21, 25, 28] as well as the predictions for the appearance of interesting topological states derived by Riberolles *et al.* from the DFT calculations and symmetry analyses [28]. Furthermore, the observed behavior in $EuIn_2As_2$ complies with the calculations by Regmi *et al.* [13] and Xu *et al.* [21], which showed that magnetic states in this material are nearly degenerate, characterized by an energy difference less than 1 meV per unit cell.

To conclude, since $EuIn_2As_2$ was predicted to be an axion insulator independent on the orientation of magnetic moments, and can be a higher order topological insulator when the magnetic moments are oriented along the hexagonal $c$ axis (which can be achieved in a rather weak magnetic field), this material can be considered an easy and versatile system for the investigation of different kinds of nontrivial topological properties within the same crystalline material.

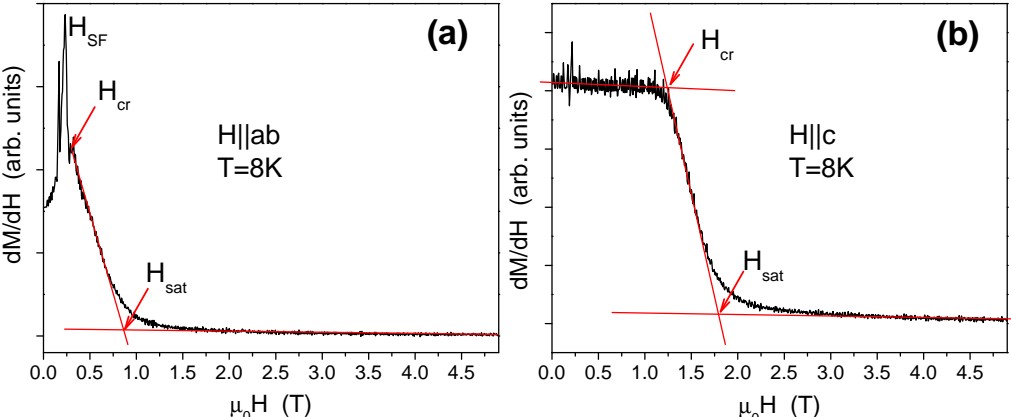

Figure 15: Definition of the critical fields in single-crystalline $EuIn_2As_2$ observed in the magnetization data taken with magnetic field applied (a) in the *a-b* plane and (b) along the *c* axis. The labels $H_{sf}$, $H_{cr}$ and $H_{sat}$ denote positions of spin-flop, antiferromagnetic and field-induced full-saturation transitions, respectively.

# Acknowledgements

**Funding information** This work was supported by the National Science Centre (Poland) under research grant 2021/41/B/ST3/01141. For the purpose of Open Access, the authors have applied a CC-BY public copyright licence to any Author Accepted Manuscript (AAM) version arising from this submission.

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
