# Peer review of "Magnetic properties of the putative higher-order topological insulator EuIn2As2"

_SciPost Physics Proceedings, doi:SciPost Phys. Proc. 11, 005 (2023)_

## Round 1 · Referee Report · Anonymous · 2022-10-22

Strengths
1- Very comprehensive characterization
2- Sound analysis and characterization
3- Impressive sample quality
4- Clearly explains how data was fitted and which assumptions were made
Weaknesses
1- Somewhat incremental knowledge increase
2- Disagreement with existing literature [Riberolles 2021] regarding the two antiferromagnetic transitions TN1 and TN2, which could perhaps be clarified by magnetization measurements with finer temperature step from 10-20 K.
Report
The authors have performed a very comprehensive study characterizing the axion insulator EuIn2As2. In their introduction, they state "in the present studies we aimed at comprehensive characterization of the thermodynamic (magnetization, heat capacity, magnetocaloric effect) and electrical transport (resistivity, magnetoresistance) properties" and they deliver all these results, thoroughly analyzed. They have performed sound data analysis which has allowed them to furthermore construct the magnetic phase diagram for the two different principal directions in EuIn2As2. I particularly appreciate their thorough explanation of data analysis procedures, fitting used and even how critical temperatures were obtained (eg. Fig. 15). All in all, the authors provide extensive and thorough characterization that will provide a valuable reference for further studies investigating the interesting topological properties of EuIn2As2.
However, I find a discrepancy with the existing literature on the compound, namely the report of Riberolles et al. (Nat. Comm. 12, 999 (2021)). The authors even mention in the penultimate paragraph of Section 4: Magnetic Properties, "Remarkably, while the neutron diffraction studies by Riberolles et al. [28] have shown two AFM transitions, at TN1 = 17.6 K and TN2 = 16.2 K, in the present study only the latter one was found" however they don't elaborate reasons why they did not observe these two AFM transitions. One of the ways Riberolles et al. was able to observe this twofold AFM transition was through magnetic susceptibility measurements; however, the authors show in Figures 2 and 4 magnetic susceptibility measurements without an inset focused on the transition itself and the possible two-step nature of the transition. In fact, in Figure 4(a) it seems that one can see the two steps in eg. 0.025 T and 0.1 T measurements. There also seems to be a weak double-peak feature present in the low-field specific heat measurements (Fig. 10) which could be the signature of these two AFM transitions observed by Riberolles et al. The authors should either satisfactorily include some explanation as to why they definitively see no evidence of two AFM transitions and how this can be substantiated in the face of the two AFM transitions clearly seen by neutron scattering, or they should revisit their data (or retake it with finer temperature steps in the vicinity of the two transitions) to determine if they are present.
Requested changes
Additionally, there are several grammatical and linguistic changes that I would recommend introducing:
In section "1 Introduction":
1 - please replace "... in the creation of new materials that elude the classic classification." with "... in the creation of new materials that elude classical classification."
2 - please replace "Its influence on the valence band has revealed the power of topology of the bands with their inversion being a base for a class of topological insulators (TI). Here, history took its toll, namely..." with "Its influence on the valence band has revealed the importance of band topology, as band inversion has become the basis for a class of topological insulators (TI). Here, history has been revived; namely, ..."
3- please replace "Hence, the role of the symmetry breaking is to remove the degeneration of the Dirac point..." with "Hence, the role of symmetry breaking is to remove the degeneracy of the Dirac point..."
4 - please replace "Topological materials keep growing interest of experimentalists and theoreticians due to a variety of the observed quantum phenomena." with "Topological materials continue to attract the interest of experimentalists and theoreticians due to the variety of observed quantum phenomena."
5- please replace "Unexpectedly it turned out that the bulk-boundary correspondence has its continuation, i.e. it is shifted to lower dimensions." with "Unexpectedly, the bulk-boundary correspondence may also be shifted to lower dimensions."
6 - please replace "... but involve novel topological physics, especially new topological invariants become relevant." with "... but involve novel topological physics, and new topological invariants become especially relevant."
7- please replace "... on the direction of the applied magnetic field, hence, in fact, depending on the magnetic order." with "... on the direction of the applied magnetic field, therefore dependent on the magnetic order."
In section "2 Experimental":
8 - Please replace "They were found stable against air and moisture." with "They were found to be stable in air and moisture."
9 - please replace "... a small fragment was crumbled from a larger piece and examined..." with "... a small fragment was crushed from a larger piece and examined..."
In section "8 Magnetic phase diagrams":
10 - In the sentence, "The neutron diffraction experiment revealed that below TN = 16.1 K..." , it is unclear if the author performed the experiment or not. I would recommend introducing the citation, as "The neutron diffraction experiment of Riberolles [28]..."
In section "9 Summary":
11 - please replace "... and to exhibit in the paramagnetic state sizeable short range interactions of ferromagnetic character observed up to about 35 K." with "... and to exhibit sizeable short range interactions of ferromagnetic character in the paramagnetic state observed up to about 35 K."
12 - Please reword the last paragraph as "To conclude, since EuIn2As2 was predicted to be an axion insulator independent of the orientation of magnetic moments, and can be a higher order topological insulator when the magnetic moments are oriented along the hexagonal c-axis (which can be achieved in a rather weak magnetic field), this material can be considered an easy and versatile system for the investigation of different kinds of nontrivial topological properties within the same crystalline material."
Additionally, for the figures:
13 - in Figure 1, it is difficult to see the information in the image of the crystal in panel (b). Please either make the information at the bottom bigger or exclude this portion, introducing the scale directly on the image.
14 - Figure 4(a), maybe include an inset for fields 0.025-0.1 T that may show the two AFM transitions at 17.6 K and 16.2 K, see the comment in the report.
15 - Figure 10, perhaps include an inset with the low-field conditions which might show a two-step peak corresponding to two AFM transitions (17.6 K and 16.2 K, see report above).

---

## Round 2 · Author Response

Answer to the Referee report on the paper:
Magnetic properties of the putative higher-order topological insulator EuIn2As2
by T. Toliński and D. Kaczorowski
We are grateful the Referee for the thorough revision of our manuscript. We applied all the suggested text corrections and they turned out to be very beneficial for our manuscript. These text changes are highlighted within the pdf file. The readability of Fig.1 has been also improved. However, the most important issue raised by the Referee have concerned the AFM transitions. He is right that at first glance one could see small inflections in magnetic susceptibility and specific heat which might indicate the presence of the transition at T_N1 = 17.6 K, apart from the well-established transition at T_N2 = 16.1 K. However, to avoid a speculative discussion, we have skipped such an analysis. In fact, detailed look to the derivatives do not allow us to state TN1 for our crystals. We have now extended Fig.4 to panels (c)-(f) showing exemplary derivatives (we have dense temperature data for the magnetic susceptibility and only for the sake of readability we skipped some curves). While for the ab plane maxima of chi(T) (corresponding to T_N2 and spin-flop T_SF) defined by zero points of derivative (T_N2 requires an extrapolation to cross the zero value of the derivative) are relatively well identified, the presence of T_N1 is not obvious. It could, maybe, be assigned to the negative peak, corresponding to a slope change because, probably accidentally, it fits T_N1. However, such a slope change is typically present above a peak of magnetic susceptibility (the slope must change somewhere) and usually does not imply a presence of any additional transition. For the c axis direction only a single transition is detected as, obviously, the spin-flop contribution is not present. Similarly, specific heat data does not allow us to claim that the transition at T_N1 is present. This is now illustrated by the inset to Fig.10. However, as we now write in the corrected manuscript, we do not definitely exclude the presence of the transition at TN1, which can be masked by overlapping anomalies associated with the various transitions at T_N1, T_N2, and T_SF. Such explanation is now added in the text, which is outlined in red.
We hope that the corrections and explanations justify our restriction to single magnetic ordering temperature.
Authors
Magnetic properties of the putative higher-order topological insulator EuIn2As2
by T. Toliński and D. Kaczorowski
We are grateful the Referee for the thorough revision of our manuscript. We applied all the suggested text corrections and they turned out to be very beneficial for our manuscript. These text changes are highlighted within the pdf file. The readability of Fig.1 has been also improved. However, the most important issue raised by the Referee have concerned the AFM transitions. He is right that at first glance one could see small inflections in magnetic susceptibility and specific heat which might indicate the presence of the transition at T_N1 = 17.6 K, apart from the well-established transition at T_N2 = 16.1 K. However, to avoid a speculative discussion, we have skipped such an analysis. In fact, detailed look to the derivatives do not allow us to state TN1 for our crystals. We have now extended Fig.4 to panels (c)-(f) showing exemplary derivatives (we have dense temperature data for the magnetic susceptibility and only for the sake of readability we skipped some curves). While for the ab plane maxima of chi(T) (corresponding to T_N2 and spin-flop T_SF) defined by zero points of derivative (T_N2 requires an extrapolation to cross the zero value of the derivative) are relatively well identified, the presence of T_N1 is not obvious. It could, maybe, be assigned to the negative peak, corresponding to a slope change because, probably accidentally, it fits T_N1. However, such a slope change is typically present above a peak of magnetic susceptibility (the slope must change somewhere) and usually does not imply a presence of any additional transition. For the c axis direction only a single transition is detected as, obviously, the spin-flop contribution is not present. Similarly, specific heat data does not allow us to claim that the transition at T_N1 is present. This is now illustrated by the inset to Fig.10. However, as we now write in the corrected manuscript, we do not definitely exclude the presence of the transition at TN1, which can be masked by overlapping anomalies associated with the various transitions at T_N1, T_N2, and T_SF. Such explanation is now added in the text, which is outlined in red.
We hope that the corrections and explanations justify our restriction to single magnetic ordering temperature.
Authors

---

## Round 2 · List of Changes

- text corrections indicated by the Referee are done and shown in red,
- the readability of Fig.1b has been improved (increased figure and scale information),
- Fig.4 is extended to additional panels (c)-(f) showing exemplary derivatives,
- Fig.10: inset added,
- Text is added (in red, page 6), which explains that we do not definitely exclude the presence of the transition at T_N1, which can be masked by overlapping anomalies associated with the various transitions at T_N1, T_N2, and T_SF.

---

## Editorial Decision

published